# COPD in Smoking and Non-Smoking Community Members Exposed to the World Trade Center Dust and Fumes

**DOI:** 10.3390/ijerph19074249

**Published:** 2022-04-02

**Authors:** Ridhwan Y. Baba, Yian Zhang, Yongzhao Shao, Kenneth I. Berger, Roberta M. Goldring, Mengling Liu, Angeliki Kazeros, Rebecca Rosen, Joan Reibman

**Affiliations:** 1Division of Pulmonary, Critical Care and Sleep Medicine, Department of Medicine, New York University Grossman School of Medicine, New York, NY 10016, USA; ridhwan.baba@nyulangone.org (R.Y.B.); kenneth.berger@nyulangone.org (K.I.B.); roberta.goldring@nyulangone.org (R.M.G.); angeliki.kazeros@nyulangone.org (A.K.); 2Division of Biostatistics, Department of Population Health and Environmental Medicine, New York University Grossman School of Medicine, New York, NY 10016, USA; yian.zhang@nyulangone.org (Y.Z.); yongzhao.shao@nyulangone.org (Y.S.); mengling.liu@nyulangone.org (M.L.); 3Department of Psychiatry, New York University Grossman School of Medicine, New York, NY 10016, USA; rebecca.rosen@nychhc.org; 4Department of Environmental Medicine, New York University Grossman School of Medicine, New York, NY 10016, USA

**Keywords:** Chronic Obstructive Pulmonary Disease, asthma, World Trade Center, September 11 terrorist attacks, dust, air pollutants, lung/physiology, New York City, spirometry, oscillometry

## Abstract

Background: The characteristics of community members exposed to World Trade Center (WTC) dust and fumes with Chronic Obstructive Pulmonary Disease (COPD) can provide insight into mechanisms of airflow obstruction in response to an environmental insult, with potential implications for interventions. Methods: We performed a baseline assessment of respiratory symptoms, spirometry, small airway lung function measures using respiratory impulse oscillometry (IOS), and blood biomarkers. COPD was defined by the 2019 GOLD criteria for COPD. Patients in the WTC Environmental Health Center with <5 or ≥5 pack year smoking history were classified as nonsmoker-COPD (ns-COPD) or smoker-COPD (sm-COPD), respectively. Main Results: Between August 2005 and March 2018, 467 of the 3430 evaluated patients (13.6%) fit criteria for COPD. Among patients with COPD, 248 (53.1%) were ns-COPD. Patients with ns-COPD had measures of large airway function (FEV1) and small airway measures (R_5–20_, AX) that were less abnormal than those with sm-COPD. More ns-COPD compared to sm-COPD had a bronchodilator (BD) response measured by spirometry (24 vs. 14%, *p* = 0.008) or by IOS (36 vs. 21%, *p* = 0.002). Blood eosinophils did not differ between ns-COPD and sm-COPD, but blood neutrophils were higher in sm-COPD compared to ns-COPD (*p* < 0.001). Those with sm-COPD were more likely to be WTC local residents than ns-COPD (*p* = 0.007). Conclusions: Spirometry findings and small airway measures, as well as inflammatory markers, differed between patients with ns-COPD and sm-COPD. These findings suggest potential for differing mechanisms of airway injury in patients with WTC environmental exposures and have potential therapeutic implications.

## 1. Background

Chronic Obstructive Pulmonary Disease (COPD) is characterized by persistent respiratory symptoms and progressive irreversible airflow limitation that develops due to airway disease and emphysema after exposure to tobacco as well as other noxious particles or gases [1]. It is increasingly recognized that COPD is a heterogeneous disease and varies in its clinical presentation, imaging, lung physiology, inflammatory markers, and prognosis [2]. The pathway of injury in COPD, such as exposure to dust, gases, and fumes, may account for some of this heterogeneity [3,4,5]. Indeed, biomass-associated COPD may differ from tobacco-related COPD [6,7]. Differences in the location of particle deposition, dose, and/or composition of inhaled toxins may contribute to the variation in disease pathobiology, presentation, or progression.

Many community members, as well as individuals involved in rescue and recovery, had acute exposure to high levels of highly irritant alkaline dust and fumes with complex components after the destruction of the World Trade Center (WTC) towers on 9/11. Many continued to have chronic exposure to resuspended dust and fumes over the ensuing months [8]. Community members included those escaping the towers, local residents, local workers, people involved in clean-up activities, as well as those passing by the WTC area on 9/11. Numerous reports now document the onset of lower respiratory symptoms of shortness of breath, cough, and wheeze in these populations [9,10,11]. Our prior data suggest that most symptomatic community members had normal spirometry; however, a subset of community members had abnormalities in small airway function and some had severe abnormalities in spirometry, which fits the criteria for COPD [12,13,14]. 

COPD has been described in WTC-exposed firefighters [15]. However, firefighters have extensive occupational exposures in addition to WTC-related exposure that differ from the acute and chronic exposures of community members. Furthermore, the study in firefighters did not differentiate between smokers and non-smokers. The WTC exposures, including interactions between tobacco smoking and WTC exposure, and their effects on COPD have not been investigated in WTC-exposed community members. The WTC Environmental Health Center (EHC) is a treatment and surveillance program for community members with WTC exposures. We now identify patients with lower respiratory symptoms and spirometry findings consistent with the GOLD (Global Initiative for Chronic Obstructive Lung Disease) definition of COPD in the WTC EHC, and compare characteristics of non-smokers with COPD (ns-COPD) to those without COPD, and to those with a history of tobacco use and COPD (sm-COPD). We examine the association of acute and chronic WTC exposures and physical and mental health symptom burden with COPD. In addition, we evaluate lung function parameters of large and small airways, including airflow reversibility, and blood-derived inflammatory biomarkers. Defining physiologic and biologic characteristics in WTC-related COPD patients may help define disease trajectory and treatment interventions, including smoking cessation and a potential focus on anti-inflammatory medications or delivery of medications to the small airways.

## 2. Methods

### 2.1. Study Population

The WTC EHC at Bellevue Hospital in New York City is a treatment and monitoring program for self-referred community members with WTC-related exposures and reported physical or mental health symptoms or cancers [8,10]. Patients undergo an initial evaluation that includes an exposure history, standardized symptom assessment, lung function measurements, and baseline complete blood counts [10]. Patients were included in this current analysis if they were ≥18 years on initial evaluation had valid pre and post bronchodilator (BD) spirometry, and were enrolled in the Bellevue Hospital program between 17 August 2005 and 30 March 2018. Patients were included as COPD if they reported lower respiratory symptoms with new onset after 9/11 and had a post BD FEV_1_/FVC < 0.7. Control patients included those without lower respiratory symptoms at the time of their enrollment in the program, normal spirometry defined by an FEV_1_, FVC, and FEV_1_/FVC greater than 0.7 and the lower limits of normal [16,17], and <5 pack year smoking history. Patients were excluded if they reported a history of wheezing before 9/11, other lung diseases including sarcoidosis, interstitial lung disease, and lung cancer, or if they were missing tobacco smoking data.

Acute WTC dust cloud exposure was defined by a report of having been caught in the dust or cloud(s) created by the collapse of the WTC buildings on 9/11. WTC-related exposure was further classified by the potential for exposure to acute and/or chronic WTC dust/fumes as a local resident, local worker, or clean-up worker based on the description of location and activities. In addition, we included information on WTC dust/ash exposure at home or in the workplace. 

New-onset lower respiratory symptoms were defined by the presence of at least one symptom of wheezing, chest tightness, or dyspnea with onset after 9/11. Symptomatic patients at initial presentation to the WTC EHC were defined based on a symptom frequency ≥2 times per week in the 4 weeks preceding enrollment in the program. Breathlessness was assessed with the modified British Medical Research Council (mMRC) dyspnea scale [18]. Standardized mental health screening was performed for PTSD symptoms (PTSD Checklist; PCL-17); a score of ≥44 was considered positive [19,20], and for depression and anxiety (Hopkins Symptom Checklist (HSCL-25), a score of ≥1.75 was considered positive [21]. 

COPD was defined by the 2019 GOLD criteria [1] as the presence of post BD FEV_1_/FVC < 0.7 measured at the first screening spirometry visit and symptoms of dyspnea, cough, wheeze in the 4 weeks preceding the initial visit to the WTC EHC. COPD patients with ≥5 and <5 pack year smoking history were classified as smoker (sm-COPD) and non-smoker (ns-COPD) COPD, respectively. We used a threshold of <5 p-y as a relatively stringent definition, but with more latitude than the standard 100 cigarettes in a life time. The Institutional Review Board of New York University Grossman School of Medicine approved the research database (NCT00404898), and only data from patients who provided written informed consent were used for analysis. 

### 2.2. Measurement of Lung Function

Spirometry and forced oscillation were routinely performed on patients before and 15 min after BD administration (2.5 mg albuterol sulfate delivered via nebulizer over 5 min) according to standard ATS/ERS guidelines [22,23]. Predicted values for spirometry measures were derived from NHANES III [16] to remain consistent with our previous publications. Data collected included FEV_1_, FVC, and FEV_1_/FVC. A BD response was defined for spirometry as an increase in FEV_1_ of 200 cc and 12%. Respiratory oscillometry was measured using impulse oscillometry (IOS) (Jaeger Impulse Oscillation System; Jaeger USA; Yorba Linda, CA, USA). Measurements were performed in accordance with European Respiratory Society recommendations [23] and as previously described [24]. Only data from trials with nearly constant tidal volume were analyzed. Reproducible tests (variability < 10%) were analyzed. Oscillometry data included measures of respiratory resistance (resistance at an oscillating frequency of 5 and 20 Hz defined as R_5_ and R_20_, respectively), frequency dependence of resistance, calculated as the difference between resistance at 5 Hz and 20 Hz (R_5–20_), and reactance area (AX), calculated as the area under the reactance curve from 5Hx to the resonant frequency. A BD response for oscillometry was defined as a reduction in R_5_ of 1.4 cmH_2_O/(L/s) [25].

### 2.3. Statistical Analysis

Statistical analyses were performed using SAS 9.4 software (SAS Institute Inc., Cary, NC, USA) and R 4.0.2 (The R Foundation). Summary statistics of univariate analyses are presented as count with percentage for categorical variables and median with IQR for continuous variables. Pearson’s Chi-squared test (or Fisher’s exact test) was used for detecting relationships between categorical characteristics. Kruskal–Wallis test was used to detect presence of an overall significant difference among multiple groups (e.g., non-COPD, ns-COPD, and sm-COPD), and Mann–Whitney U test was used to detect differences between two groups for continuous variables. The presence of symptoms suggestive of COPD and abnormal pulmonary function tests was studied in a logistic regression model adjusted for age, gender, and race. Two-tail *p*-values < 0.05 were considered statistically significant.

## 3. Results

### 3.1. Demographic and Exposure Characteristics of COPD Patients in the WTC EHC

Among patients in the Bellevue Hospital WTC EHC, after exclusion as defined, 3430 patients fit the criteria for inclusion in the study (Figure 1). 

Among these patients, 467 patients (13.6%) fit criteria for COPD. As shown (Table 1), patients with COPD were significantly older than those without COPD (median age 61 vs. 52 years, respectively), more likely to be male, and had significant differences in race/ethnicity. We did not observe a difference in BMI. We did not observe significant differences in acute WTC exposure (WTC dust cloud exposure) between the COPD and non-COPD groups (*p* = 0.36). There were slight differences in exposure categories between COPD and non-COPD, and patients with COPD were more likely to report WTC ash in the home and to have been involved in cleaning their home compared with the non-COPD (data not shown). 

We compared the ns-COPD patients (*n* = 248) with ns-control (*n* = 2361) (Table 1). Nearly 10% of the non-smoking WTC EHC patients fit the definition of COPD. As shown, ns-COPD were slightly older and included more men. Differences in race/ethnicity were also noted between the ns-COPD and ns-control. We did not detect a difference in acute dust exposure (WTC dust cloud), exposure category, and home or work exposure between these two groups. Few patients reported non-WTC-related exposure to dust as an occupation or hobby. 

We further compared ns-COPD patients in the WTC EHC with sm-COPD (Table 1) to understand whether the pathways to COPD were associated with different patient characteristics. Patients with ns-COPD compared to sm-COPD were slightly younger (median age 58 vs. 62 respectively, *p* < 0.001). Compared to ns-COPD, sm-COPD were more often local residents (*p* = 0.007), and reported greater WTC ash exposure in the home (*p* = 0.005). Reported respiratory and mental health symptoms were similar between ns-COPD and sm-COPD (data not shown). 

### 3.2. Lung Function in ns-Control, ns-COPD, and sm-COPD in the WTC EHC

In Table 2, we show spirometry values for ns-Control, ns-COPD, and sm-COPD. By definition, patients with COPD had reduced post BD spirometry measures compared with ns-Control. Compared to ns-Controls, ns-COPD were more likely to have a significant BD response defined by spirometry (23.8% vs. 5.0%, *p* < 0.001).

We further compared spirometry values in ns-COPD and sm-COPD, including the presence of BD reversibility. Both ns-COPD and sm-COPD had mild–moderate COPD by spirometry criteria (Table 2). We did not identify any difference in spirometry values of pre BD FVC, FEV_1_, or FEV_1_/FVC between the two groups. There was a slight difference in post BD FEV_1_/FVC, with a higher value in the ns-COPD. Although many patients failed to have a defined BD response, more ns-COPD compared to sm-COPD patients had a BD response as defined by change in FEV_1_ (23.8% vs. 14.1%, *p* = 0.008). 

We also used oscillometry to assess small airway function and BD responsiveness. Oscillometry measures in both ns-COPD and sm-COPD groups were significantly more abnormal than ns-Control (Table 2). More patients with ns-COPD compared with ns-Controls had a BD response by oscillometry (35.5% vs. 13.3%, *p* < 0.001). Median Pre BD measures of R_5_, R_5–20_, and Ax were elevated in both ns-COPD and sm-COPD, but no differences beween the two groups were identified. Both groups had a slight decrease in frequency dependence of resistance and reactance values (R_5–20_ and AX) with BD; however, ns-COPD had lower values after BD compared to sm-COPD, consistent with a greater improvement in small airway function. When assessed by examining the number of patients who had a standardized BD response measured by IOS, more ns-COPD patients had a BD response defined by change in R5 compared to sm-COPD (35.5% vs. 21.3%, *p* = 0.002). 

To confirm that results were not influenced by confounders, we adjusted symptoms and lung function including oscillometry results for covariates identified as significant in our univariate analysis (age, gender, and race/ethnicity; Table 3). Post BD FEV_1_ and post BD R_5_, R_5–20_ and Ax remained significantly different between ns-COPD and sm-COPD.

### 3.3. Blood Biomarkers in ns-Control, ns-COPD, and sm-COPD in the WTC EHC

Clinical blood biomarkers were obtained from patients upon enrollment in the WTC EHC. These included complete blood counts with differential. Patients with ns-COPD had significantly higher blood eosinophil counts compared with ns-control (*p* = 0.001), although levels were only minimally elevated. These findings were also noted if we used % of eosinophils or neutrophils (data not shown). No difference in blood neutrophils was observed between ns-COPD and ns-Control (Table 4). We did not identify any differences between blood eosinophil counts between ns-COPD and sm-COPD. In contrast, sm-COPD had a higher absolute median blood neutrophil count compared with ns-COPD (Table 4).

## 4. Discussion

There is a growing focus on environmental factors in addition to tobacco smoke as risks for COPD. Analysis of the WTC population with a non-occupational defined environmental exposure and without a history of tobacco use has the potential to inform these studies. We examined patients in the WTC EHC with COPD defined by GOLD criteria to further our understanding of the role of an environmental exposure on the development of spirometry-defined COPD. We identified differences in pulmonary physiology, including bronchodilator response and inflammatory biomarkers in ns-COPD compared to sm-COPD. Our data are consistent with heterogeneity within COPD and suggest differences between WTC exposure and tobacco smoking-related COPD. 

We defined COPD by the GOLD 2019 guidelines, and report a rate of 13.6% of the patients in the WTC EHC who fit criteria for COPD. Of the non-smokers, 9.6% fit criteria for ns-COPD. This prevalence of COPD in non-smokers is higher than the rate of 2.2–6.5% reported in studies of those in different occupations [26], in non-smokers [27], or in the Multi-Ethnic study of Atherosclerosis (MESA) [28]. The rate is consistent with the influence of occupational exposures on COPD, which are estimated to account for 15% of the disease [29]. The high rate of ns-COPD in our population may be due in part to recruitment bias, since patients were required to report any of a variety of symptoms in order to enroll in the WTC EHC, and most of these symptoms were respiratory symptoms; however, the rate is consistent with the 12% of symptomatic non-smokers with COPD described in the Copenhagen General Population study [30]. This high rate provided us with a sizeable population to characterize differences among those with and without COPD and between ns-COPD and sm-COPD. 

We defined our patients as COPD based on the presence of lower respiratory symptoms with persistent abnormality in airflow on post BD spirometry. However, there is well-described overlap between definitions of asthma and COPD [1,30,31,32] and a lack of international consensus on the definition and diagnostic criteria for asthma-COPD overlap. Clinical features between asthma and COPD often overlap, despite attempts to differentiate the two diseases [33]. The 2018 Asthma Lancet Commission [34] suggests that clinical or inflammatory characteristics and underlying mechanisms should be used to classify asthma and COPD to support personalized management and improve clinical outcomes. Thus, we characterized our patients by clinical characteristics, physiology, and blood biomarkers. 

We identified some clinical and demographic differences between COPD patients and those without COPD. Patients with COPD were older, more likely to be male, and more symptomatic. By definition, they had reduced lung function compared to the control population. Surprisingly, we did not identify clear differences among our groups for acute WTC exposure, as defined by dust cloud exposure. There was a suggestion that chronic exposure, as defined by WTC ash in the home or being a local resident was associated with sm-COPD, although we could not identify this as a risk for ns-COPD. This finding suggests potential interplay between tobacco smoking and WTC dust in residents (being local resident and with WTC ash at home).

We identified some important differences in lung physiology between the ns- and sm-COPD. Spirometric measures of pre BD FEV_1_, FVC, and FEV_1_/FVC were similar among the ns-COPD and sm-COPD; however, post BD FEV1/FVC was relatively preserved in ns-COPD compared with sm-COPD. Moreover, more ns-COPD patients had a BD response assessed by spirometry compared to sm-COPD. We used oscillometry to assess small airway function including measures that reflect total resistance (R_5_), as well as non-uniformity of small airway function (R_5–20_, AX). Both ns-COPD and sm-COPD had abnormal measures of small airway function without differences between ns-COPD and sm-COPD at baseline. However, we observed a significant improvement in small airway measures in ns-COPD compared to sm-COPD. In addition, more ns-COPD fit criteria for a BD response measured by oscillometry compared to sm-COPD. Importantly, post BD values of FEV_1_ and oscillometry measures remained significantly improved in ns-COPD compared with sm-COPD even after adjusting for potential confounders. We have demonstrated the importance of BMI in predictive equations for measures of respiratory impedance [35]; however, we did not identify a difference in median BMI across our three groups. Nevertheless, there was residual small airway dysfunction in both groups, suggesting potential for a fixed airway defect. Of importance, our prior data demonstrated that oscillometry can detect the presence of both a reversible and an irreversible small airway injury [24]. Thus, the enhanced small airway abnormality seen in sm-COPD compared with ns-COPD suggests presence of an airway injury due to smoking in addition to exposure to WTC dust, which may reflect differences in airway inflammation or structure between the two groups.

To evaluate for differences in inflammation between ns-COPD and sm-COPD, we examined markers of inflammation using blood eosinophil and neutrophil counts. Blood eosinophils were slightly increased in ns-COPD compared with ns-control. As such, this group may have similarities to patients with asthma with persistent or fixed airflow limitation [36,37,38,39,40,41]. This irreversible or partially reversible airflow obstruction in asthma develops from airway remodeling secondary to airway inflammation, epithelial damage, abnormal tissue repair, angiogenesis, and/or increase in airway smooth muscle mass and although often associated with prolonged asthma, may be a response to many risk factors [31]. We did not clearly identify a difference in blood eosinophils between the ns-COPD and sm-COPD. This finding is similar to the recent NOVELTY cohort, in which blood eosinophils did not differentiate between Asthma, Asthma + COPD, and COPD in those patients with physician-assigned diagnosis [42]. We did find an increase in peripheral blood neutrophils in those with sm-COPD compared to ns-COPD. This finding suggests the potential for differences among inflammatory pathways in these groups with COPD and may point to different pathways of airway remodeling or structural changes.

There are some limitations to this study. This is a retrospective case control study with patients self-referring to a treatment program. Although enrollment is ongoing and there is a potential that enrollment period might influence the lung function measure, we did not see a major difference in the distribution in years of enrollment between the three groups of study. We monitored patients that did not have preexisting history of lung disease and measurement of lung function prior to their WTC exposures, and although we excluded patients with pre-existing respiratory symptoms, the possibility exists that some of these patients may have had lung disease or asthma prior to their WTC exposures and other factors that can influence the childhood development of lung disease [4,43]. Although occupational exposures to vapor, gas, dust, and fumes can contribute to the development of COPD [5,28], we have not analyzed these exposures in detail, and used WTC dust/fume exposure as a single environmental exposure. We did not identify any difference in reports of known exposures to dust from work or hobbies between the three groups. Furthermore, we do not have measures of allergen sensitivity, which might be associated with an atopic asthma phenotype. Abnormal findings, including emphysema on CT scans, are increasingly being reported in patients with lower respiratory symptoms and COPD. We do not have consistent CT studies in these patients; however, intriguingly, findings of emphysema [44] and increased airway wall thickness [45] have been reported in WTC workers. We only used a history of tobacco use and did not measure cotinine levels to test for current self-or environmental tobacco exposure. Finally, the longitudinal trajectories of ns-COPD and sm-COPD may differ and deserve future study.

In summary, our study suggests that COPD in non-smoking patients with WTC exposure has physiologic findings that are similar to those with a history of tobacco use; however, we suggest some important differences. Both sm-COPD and ns-COPD groups have evidence of airway injury throughout the airway tree, including small airway abnormalities. More patients with ns-COPD display bronchodilator responsiveness as measured by both spirometry or oscillometry, suggesting some airflow reversibility throughout the airway tree in the ns-COPD compared to sm-COPD. Although blood eosinophils did not discriminate between ns- and sm-COPD, we identified an increase in blood neutrophils in patients with sm-COPD consistent with different mechanisms of injury. Furthermore, we suggest a potential interplay between tobacco smoking and residential exposure to WTC dust and fumes with COPD. These data have potential implications for differing mechanisms of disease and treatment. Longitudinal assessment of symptoms and lung function, as well as more detailed analysis of WTC exposures, will further our understanding of lung injury leading to COPD in a non-smoking population, as well as the interaction between environmental exposures. 

## Figures and Tables

**Figure 1 ijerph-19-04249-f001:**
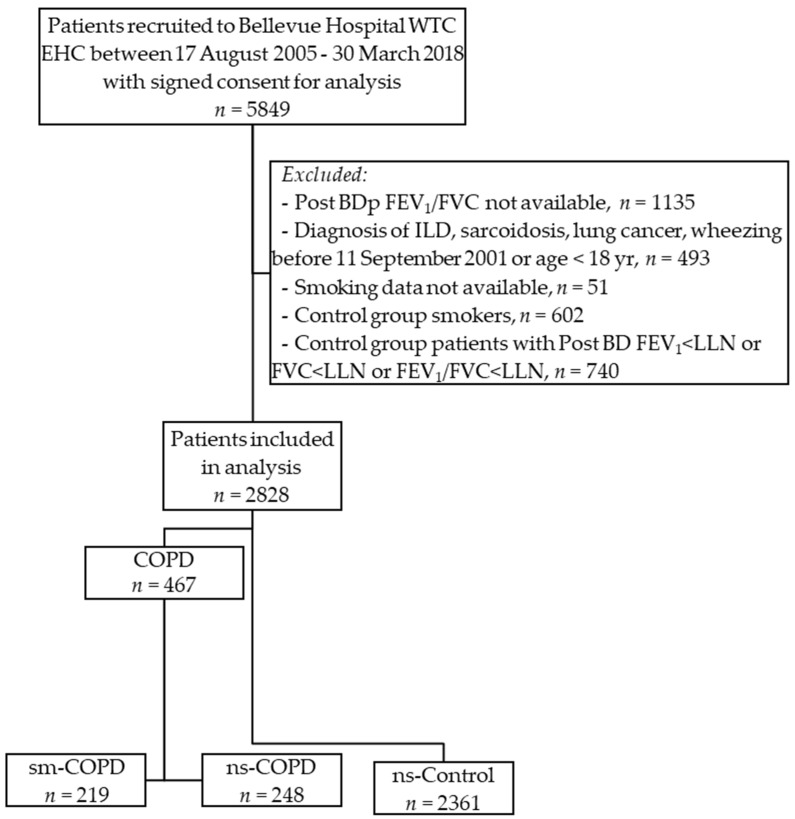
Inclusion and exclusion criteria for the study.

**Table 1 ijerph-19-04249-t001:** Characteristics of patients in the WTC EHC.

	ns-Control (*n* = 2361)	ns-COPD (*n* = 248)	sm-COPD (*n* = 219)	*p*-Value ^1^	*p*-Value ^2^
9/11 to enrollment, median (IQR) in years	8.8 (5.9)	9.4 (6.9)	9.7 (6.3)	0.20	0.56
Age, median (IQR) in years	51 (15)	58 (14)	62 (13)	<0.001	<0.001
Male, *n* (%)	1116 (47)	159 (64)	146 (67)	<0.001	0.56
Female, *n* (%)	1245 (53)	89 (36)	73 (33)		
Race/Ethnicity, *n* (%)				<0.001	0.13
White	931 (40)	130 (53)	138 (63)		
Hispanic	694 (30)	42 (17)	39 (18)		
Black	519 (22)	52 (21)	30 (14)		
Asian	178 (8)	19 (8)	9 (4)		
Others	19 (1)	3 (1)	2 (1)		
Education, *n* (%)				0.41	0.39
Up to 6th grade	174 (7)	17 (7)	14 (6)		
High school and higher	2184 (93)	231 (93)	205 (94)		
BMI, median (IQR) in kg/m^2^	27.4 (7.1)	27.3 (7.1)	26.8 (7.6)	0.67	0.40
BMI, *n* (%)				0.45	0.31
Underweight (0–18.5)	26 (1)	5 (2)	11 (5)		
Normal (18.5–24.9)	681 (29)	73 (30)	63 (29)		
Overweight (25.0–29.9)	874 (38)	83 (34)	76 (35)		
Obese (≥30)	743 (32)	83 (34)	66 (31)		
WTC Exposure Category, *n* (%)				0.13	0.007
Local worker	1322 (56)	136 (55)	116 (53)		
Resident	493 (21)	51 (21)	70 (32)		
Rescue/Recovery/Other	286 (12)	41 (17)	20 (9)		
Clean-up worker	254 (11)	19 (8)	11 (5)		
WTC dust cloud exposure, *n* (%)	1255 (54)	140 (57)	122 (56)	0.35	0.88
WTC ash exposure, *n* (%)					
At home	426 (18)	49 (20)	68 (31)	0.52	0.005
Cleaned home	385 (16)	49 (20)	60 (28)	0.17	0.05
At workplace	1069 (60)	114 (61)	97 (58)	0.76	0.59
Cleaned workplace	775 (43)	82 (44)	74 (43)	0.83	0.91
Non-WTC-related dust exposure, *n* (%) *				0.10	0.62
Yes	110 (7)	19 (10)	14 (8)		
No	1589 (94)	173 (90)	164 (92)		

^1^ based on comparison tests between ns-COPD and ns-control; ^2^ based on comparison tests between ns-COPD and sm-COPD; * There are some missing values for this past occupation or hobby-related dust exposure question since the question was added at a later date: 28% missing in ns-control, 23% missing in ns-COPD, and 19% missing in sm-COPD.

**Table 2 ijerph-19-04249-t002:** Spirometry and oscillometry results *.

	ns-Control (*n* = 2361)	ns-COPD (*n* = 248)	sm-COPD (*n* = 219)	*p*-Value ^1^	*p*-Value ^2^
Spirometry Median (IQR)					
Pre BD-FEV1 (% predicted)	96.1 (15.9)	73.7 (23.3)	71.8 (25.7)	<0.001	0.51
Pre BD-FVC (% predicted)	96.6 (15.8)	91.5 (27.6)	89.4 (25.7)	<.0001	0.9
Post BD-FEV1 (% predicted)	99.4 (16.1)	77.7 (23.8)	75.9 (23.5)	<0.001	0.08
Post BD-FVC (% predicted)	96.2 (15.4)	95.8 (25.9)	93.4 (24.3)	0.012	0.83
Post BD-FEV1/FVC	0.82 (0.07)	0.66 (0.07)	0.63 (0.14)	<0.001	<0.001
Bronchodilator response *n* (%)	118 (5)	59 (23.8)	31 (14.1)	<0.001	0.008
Forced Oscillation Median (IQR)					
Pre BD-R_5_ (cmH_2_O/[L/s])	4.4 (2.2)	5.5 (2.9)	5.4 (2.6)	<0.001	0.58
Pre BD-R_5–20_ (cmH_2_O/[L/s])	0.9 (1.0)	1.6 (1.9)	1.6 (1.8)	<0.001	0.99
Pre BD-AX (cmH_2_O/L)	5.1 (8.1)	12.8 (23.6)	13.8 (23.6)	<0.001	0.39
Post BD-R_5_ (cmH_2_O/[L/s])	3.8 (1.9)	4.3 (2.3)	4.6 (2.4)	<0.001	0.18
Post BD-R_5–20_ (cmH_2_O/[L/s])	0.7 (0.8)	1.0 (1.2)	1.1 (1.5)	<0.001	0.04
Post BD-AX (cmH_2_O/L)	3.3 (5.1)	5.2 (10.8)	7.3 (15.3)	<0.001	0.01
Bronchodilator response *n* (%)	264 (13.3)	77 (35.5)	41 (21.3)	<0.001	0.002

^1^ based on comparison tests between ns-COPD and ns-control using Mann–Whitney test; ^2^ based on comparison tests between ns-COPD and sm-COPD using Mann–Whitney test; * Kruskal–Wallis test for overall difference among the three groups is significant for all markers (*p* < 0.005).

**Table 3 ijerph-19-04249-t003:** Symptoms, spirometry, and oscillometry adjusted by age, gender, and race using logistic regression models.

	ns-COPD vs. ns-Control		sm-COPD vs. ns-COPD	
	OR	*p*-Value	OR	*p*-Value
Symptoms within last 4 weeks				
Cough	1.776	<0.001	0.806	0.29
Wheezing	3.246	<0.001	1.091	0.66
Chest tightness	1.916	<0.001	0.827	0.34
Dyspnea with exercise	2.391	<0.001	1.232	0.4
Dyspnea at rest	2.025	<0.001	1.205	0.36
mMRC score ≥ 3	1.977	<0.001	0.858	0.51
Spirometry				
Pre BD-FEV1	0.874	<0.001	0.994	0.18
Pre BD-FVC	0.962	<0.001	1.001	0.87
Post BD-FEV1	0.873	<0.001	0.989	0.02
Post BD-FVC	0.984	0.003	1.000	0.98
Forced Oscillation				
Pre BD-R_5_	1.417	<0.001	1.034	0.5
Pre BD-R_5–20_	2.099	<0.001	1.065	0.41
Pre BD-Ax	1.071	<0.001	1.007	0.20
Post BD-R_5_	1.315	<0.001	1.156	0.012
Post BD-R_5–20_	1.836	<0.001	1.336	0.005
Post BD-Ax	1.074	<0.001	1.026	0.004

**Table 4 ijerph-19-04249-t004:** Blood biomarkers in ns-Controls, ns-COPD, and sm-COPD *.

	ns-Control (*n* = 2361)	ns-COPD (*n* = 248)	sm-COPD (*n* = 219)	*p*-Value ^1^	*p*-Value ^2^
Blood biomarkers					
Eosinophil count Median (IQR) cells/microL	117.6 (130)	135.0 (176)	158.4 (165)	0.001	0.59
Neutrophil count Median (IQR) cells × 1000/microL	3.9 (1.9)	3.8 (1.7)	4.7 (2.6)	0.92	<0.001

^1^ based on comparison tests between ns-COPD and ns-control using Mann–Whitney test; ^2^ based on comparison tests between ns-COPD and sm-COPD using Mann–Whitney test; * Kruskal–Wallis test for overall difference among the three groups is significant for all markers (*p* < 0.005).

## Data Availability

De-identified data, including a limited data dictionary, will be made available to investigators in the general scientific community that request information. The Research Oversight Committee of the WTC EHC will review all requests and make a judgment based on the affiliation with a scientific or educational institution, and on the basis of the reason for the request. The data will be made available for two years after publication.

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
