# Peer review of "COPD in Smoking and Non-Smoking Community Members Exposed to the World Trade Center Dust and Fumes"

_ijerph, 2022, doi:10.3390/ijerph19074249_

Round 1

Reviewer 1 Report

Summary. The authors used spirometry and impulse oscillometry (IOS) to assess lung function and World Trade Center (WTC)-related COPD in 3,430 persons in the Lower Manhattan community who survived the WTC attacks of Sept. 11, 2001, of whom 467 were diagnosed with post-WTC COPD. Those with COPD were further divided into smokers and non-smokers; those two groups differed with respect to airway measures and bronchodilator response but not with blood neutrophils. Different environmental vs. smoking-related mechanisms for development of COPD are suggested.

Comments. These are unique and valuable data from a group that has largely been the clinical champion of community members exposed to WTC, and one of the only two that have published extensively on the impact of WTC on community members’ health (the other is DOHMH, which has presented mostly questionnaire-based findings and only a small amount of pulmonary function data).

The data are clearly described and presented; I have little to say about the findings themselves. However, both in their abstract and at the end of the Background section the authors allude to “potential implications for interventions” without making specific recommendations. This somewhat leaves the reader up in the air, particularly because it implies that there are two types of COPD – the smoking-related kind and the environmentally induced kind. What do they expect clinicians to do with this information?

There are two important methodological issues that need further comment. First, eligibility for this study is not defined well, certainly not to the extent featured in the many previous publications from this group. Of particular concern is the fact that persons could have enrolled as late as March, 2018, more than 16 years after 9/11. If the great majority entered the study in the early post-9/11 years, it’s probably not worth discussing, but a lot can happen in 16 years, including diagnosis with many other smoking-related conditions, including heart disease, and treatment which may or may not be effective, to say nothing of survival bias. This renders the median age given in Table 1 uninterpretable. At the very least Table 1 needs to show the distribution of year of enrollment.

This brings up a related question: was pulmonary function measured exactly once for each study participant? If so, does the time since WTC exposure affect the findings? If not (i.e., if there are multiple measures per person), which measurement was used and why?

The second issue is the limited treatment of occupation. There is a brief statement (line 160) that there was no difference in work exposure between the non-smoking cases and controls, plus an acknowledgment (line 218) of “growing focus on environmental factors in addition to tobacco smoke as risks for COPD.” The scant treatment of COPD in occupational medicine literature has been a topic of discussion for quite some time, including for example, the chapter on chronic airways disease by John Balmes in Rom’s textbook Environmental and Occupational Medicine, 4th Ed. (Dr. Reibman is co-author of a chapter on asthma). The authors suggest (lines 298-301) that they have additional occupational exposure data that has not been analyzed. I would think this is an ideal place for such an analysis. It might show nothing at all if few of the community members who participated in the Bellevue studies have occupations that are associated with COPD (or asthma, for that matter). Still, it would be useful to describe exactly what is known about study participants’ past occupations and potential exposures.

There are a few minor points to be addressed.

  1. Line 40 there is either a missing word or the “a” before “heterogeneous” should be deleted.

  1. Pulmonary function norms were taken from NHANES-III. These data are 20 years old by now, and while they do include a Hispanic/Latino faction, it is Mexican-American. To what extent do these norms apply to New York City Latinos who are largely Puerto Rican and Dominicano?

  1. Speaking of norms, what norms were used for IOS?

Reviewer 2 Report

  1. The onset time after 9/11 doesn't mean WTC dust exposure must be the etiological risk of COPD. The sex difference of COPD doesn't support the conclusion of this study. There was no exposure association analysis in the study either.
  2. COPD is closely associated with smoking and genetic susceptibility.  Logistic analyses should control both of the two risks and the exposure levels of WTC dust.
  3. The title doesn't match the context of this study.

Round 2

Reviewer 2 Report

No convincing evidence had been supplied to exclude the confounding effects of smoking and genetic susceptibility on COPD.